# Intensity of a Physical Exercise Programme Executed through Immersive Virtual Reality

**DOI:** 10.3390/healthcare11172399

**Published:** 2023-08-26

**Authors:** Frano Giakoni-Ramírez, Andrés Godoy-Cumillaf, Paola Fuentes-Merino, Claudio Farías-Valenzuela, Daniel Duclos-Bastías, José Bruneau-Chávez, Eugenio Merellano-Navarro, Ronald Velásquez-Olavarría

**Affiliations:** 1Facultad de Educación y Ciencias del Deporte, Instituto del Deporte y Bienestar, Universidad Andres Bello, Santiago 7550000, Chile; frano.giakoni@unab.cl; 2Grupo de Investigación en Educación Física, Salud y Calidad de Vida, Facultad de Educación, Universidad Autónoma de Chile, Temuco 4780000, Chile; paola.fuentes@uautonoma.cl (P.F.-M.); ronald.velasquez@cloud.uautonoma.cl (R.V.-O.); 3Facultad de Ciencias para el Cuidado de la Salud, Universidad San Sebastián, Lota 2465, Providencia 7510157, Chile; claudio.farias@uss.cl; 4Escuela de Educación Física, Pontificia Universidad Católica de Valparaíso, Valparaíso 2374631, Chile; daniel.duclos@pucv.cl; 5IGOID Research Group, Physical Activity and Sport Science Department, University of Castilla-La Mancha, 45071 Toledo, Spain; 6Departamento de Educación Física, Deportes y Recreación, Universidad de la Frontera, Temuco 4811230, Chile; jose.bruneau@ufrontera.cl; 7Department of Physical Activity Sciences, Faculty of Education Sciences, Universidad Católica del Maule, Talca 3530000, Chile; emerellano@ucm.cl

**Keywords:** accelerometer, light, moderate, vigorous, MVPA

## Abstract

Evidence suggests that moderate to vigorous physical activity (MVPA) is necessary for health benefits. Immersive virtual reality is a technology that uses images, sounds, and tactile sensations from a simulated world to encourage healthy behaviours and physical activity. The aims of this research are (1) to determine the duration and intensity of physical activity performed through immersive virtual reality; (2) to determine differences in physical activity intensity according to gender. Methods: A nonprobabilistic convenience sample composed of 39 university students of physical education pedagogy, who performed, through immersive virtual reality, a physical activity programme composed of three levels that progressively increased in complexity. Physical activity intensity was measured using accelerometers. Results: Of the three levels, the most complex was not the one that produced the most minutes of MVPA. The three levels added up to 08:53 min of MVPA. No significant differences were found when comparing them by sex. Conclusions: The results of this study suggest that an exercise programme delivered through immersive virtual reality generates MVPA levels, with no major differences between sexes. Further research is needed to confirm the contribution of immersive virtual reality to physical activity.

## 1. Introduction

Any muscle-generated movement that results in energy expenditure is considered physical activity (PA) [1], but there is evidence that the intensity of PA needs to be moderate to vigorous (MVPA) to achieve greater health benefits, such as preventing overweight and obesity, reducing risk factors for chronic disease, or improving the immune system [2,3]. For this reason, the WHO is developing guidelines that quantify the necessary activity (amount, frequency, and intensity) that people should perform to achieve health benefits [4,5]. Specifically, it is recommended that adults aged 18–64 years should engage in at least 150–300 min of moderate-intensity aerobic physical activity, or 75–150 min of vigorous-intensity aerobic physical activity, or an equivalent combination of moderate- and vigorous-intensity physical activity, per week to strengthen the cardiorespiratory, muscular, and skeletal systems [6].

Despite efforts to increase levels of physical activity in the population and the scientific evidence of its health benefits, only one in four adults achieves the recommended levels of physical activity, and more than 80% of adolescents worldwide are insufficiently active [7]. Among university students, the impact of physical inactivity on student health is a major concern, with the influence of sex and other determinants being identified. High mortality due to physical inactivity contributes to productivity losses [8]. The costs associated with this situation have been estimated and pose a problem because of the significant expenditure incurred by the health systems of each country, mainly by the public system but also by households [9]. Other studies have linked poor health habits that are maintained over time to dropping out of university [8].

Experts point to the need to develop strategies that contribute to the promotion of healthy lifestyles and have a global impact on the whole population, which requires a systemic approach that improves social, cultural, economic, and environmental factors [10]; however, the effectiveness of and compliance with these initiatives are not always ideal and are influenced by factors such as a lack of personal motivation, time availability, or fatigue after long working days, or simply because PA is perceived as boring or difficult [11,12,13,14].

This situation has led to the search for tools that people find attractive and that are adapted to their own interests and motivations. One such tool is video games. However, much of the evidence on the subject suggests that those who play video games frequently are more likely to have poorer physical and mental health parameters, mainly affected by lower levels of physical fitness, changes in weight due to prolonged sedentary behaviour, and poor eating habits, compared with their peers who play them less often [15]. Other studies have tried to support the position that these devices could contribute to the promotion of healthy habits, and some research has even shown that the use of these devices has various benefits for people’s health [16,17]. Therefore, it is important to consider the potential benefits as they have become increasingly complex, realistic, and social in nature [16].

One of the main strengths of video games is that they allow people to be more motivated to engage in PA, allowing them to perform actions that would not be possible, for reasons such as time or space [18]. Technological advances have increased the range of experiences and possibilities of games, and new types of video games have emerged that allow for player interaction [19]. These were followed by virtual trainers aimed at practising PA, known as exergames [20].

Technological advances have enabled the development of virtual reality (VR) devices, which artificially create sensory experiences that allow the user to interact with objects in a virtual environment [21]. Immersive virtual reality (IVR) is a type of VR characterised as a technique in which the user receives images, sounds, and tactile sensations from a simulated world using a head-mounted display [22,23]. There is evidence to support IVR as a tool to promote healthy behaviours and physical activity and to increase long-term exercise adherence [24], which justifies the increasing use of immersive virtual reality (IVR) as an innovative technique to encourage physical exercise and active behaviour [25,26,27]. IVR allows the movements of a virtual avatar to be displayed in a way that is consistent with the subject’s body in a first-person perspective, which can elicit measurable motor and physiological responses in the real body. For example, participants adjust their real actions in response to the observed movements of the avatar, driven by the illusory sensation of having the avatar’s body [28].

A recent scoping review aimed to identify publications using IVR to promote physical activity and found that there is interest in the scientific community in the practice of PA using IVR, positioning it as a method that may be more effective in developing and maintaining a healthy lifestyle, and concluding that its use stimulates MVPA levels, which could contribute to meeting physical activity recommendations [29]. When looking at the tools used to determine PA intensity in the IVR, all the studies included in the scoping review did so by monitoring heart rate (HR) [25], which is a cardiorespiratory parameter used to assess PA intensity [30], but studies have shown that the use of VR can increase HR levels. Orman et al. [31] showed that those who participated in a VR experience had higher-than-average HR values, while Varela-Aldás et al. [32] reported that, in a group of athletes exercising on a treadmill with an IVR device, their HR was lower when provided with a quiet environment and higher when provided with an exciting and threatening environment. The reason for these differences may be explained by the anxiety or claustrophobia generated in a virtual environment [33].

The tool that can objectively measure sedentary time and the amount and intensity of PA is the accelerometer [34,35]. This is a small device that, when worn by a person, measures the acceleration of the body caused by its movement [36]. Only two studies have examined the measurement of PA intensity using accelerometers during IVR activities, with the first concluding that the use of these devices can lead to varying levels of PA [37] and the second concluding that these devices lead to MVPA [38], both of which highlight the need for further research to describe the potential benefits of IVR on PA levels. For these reasons, the present study aims to (1) determine the time and intensity of physical activity performed through immersive virtual reality and (2) determine the differences in the intensity of physical activity according to sex.

## 2. Materials and Methods

### 2.1. Study Design

This is an observational, descriptive, cross-sectional study, with a quantitative research approach, carried out in the first semester of 2023, with students of Physical Education at the Universidad Autónoma de Chile, in the city of Temuco, Chile.

### 2.2. Participants

The sample for the study was a nonprobability convenience sample. The 200 students in the programme were invited to participate. The final sample consisted of 39 students (18 women and 21 men), aged between 18 and 24, who were in the second or third year of their vocational training. Inclusion criteria included voluntary participation through an informed consent document and no physical or visual impairments that would prevent them from performing an exercise programme via IVR. Participants who did not attend within the time allotted for the programme were excluded.

The research had the support of the authorities of the Directorate of the Physical Education Pedagogy Course, as well as the approval of the Scientific Ethics Committee of the Universidad Autónoma de Chile (CEC-N°42-22), which protected the anonymity of the study participants and the confidentiality of the information collected, only for educational research purposes.

### 2.3. Instruments

An Oculus Meta Quest 2 (Meta Platforms, Inc., Menlo Park, CA, USA) was used for the IVR, along with two touch controls.

The exercise programme was delivered using the FitXR (version 3.7.88) application (developed by FITXR LIMITED, London, UK), following methods described in previous research [39,40]. The FitXR application was chosen because it is easy to use for people with no previous exposure to IVR [39]. The Combat Studio version of FitXR was used, which is a martial-arts-inspired workout. The touch controls simulate boxing gloves, and combinations of punching spherical objects are performed, as well as avoiding being hit by blocks by performing squats and sideways movements. Three levels were used, which became more complex according to the speed of execution of the movements. The lowest level was called Beginner and lasted 8 min; the next level was Intermediate and lasted 9 min; the most complex level was Advanced and lasted 8 min.

Sedentary behaviour and physical activity were measured using ActiGraph wGT3X-BT accelerometers (AG; ActiGraph, Pensacola, FL, USA) during the 3 levels of the exercise programme. Subjects wore the device on the right side of the hip. Data were collected at 100 Hz. Data were then downloaded using Actilife 6.13.4 (ActiGraph, Pensacola, FL, USA) with a 10 s epoch. The cut-off points used to determine sedentary behaviour and physical activity intensities were the following: sedentary time ≤ 100 cpm, light PA = 100–1951 cpm, moderate PA = 1952–5724, and vigorous PA ≥ 5725, as proposed by Arias-Palencia et al. [41] and already used by university students [42].

### 2.4. Procedure

Data were collected between April and May 2023. University students enrolled in the Faculty of Education were invited to participate through institutional emails explaining the aims of the study, the activities to be carried out, their duration, the clothing required for their development, and the ethical rules for the treatment of the data collected.

Each participant completed the exercise programme once. Before the intervention began, the research team placed the accelerometer on the right side of each participant’s waist and explained its benefits and use. This was followed by a 15 min warm-up consisting of oxygenation, joint mobility, and stretching, all using the FitXR application. The physical exercise programme was then developed through the FitXR application, using the Combat Studio version, starting with the Beginner level (8 min), followed by the Intermediate level (9 min), and ending with the Advanced level (8 min). There was a 5 min break between each level. A cooldown activity was performed, which consisted of 10 min of stretching exercises. The total intervention time for each participant was approximately one hour.

PA intensity was measured throughout the development of the activity. For data analysis, the data collected from the beginning of the Beginner level to the end of the Advanced level were used, without considering the breaks between levels.

### 2.5. Statistical Analysis

Normality of the data was measured using the Shapiro–Wilk test. Differences between the sexes were determined using the Student’s *t*-test. Effect size was measured using the Cohen’s d test. Statistical significance was determined for all tests using the conventional *p*-value of *p* < 0.05. Data were analysed using the SPSS statistical package, version 28.

## 3. Results

The sample consisted of 39 university students, 54% of whom were men. Table 1 shows the mean values of sedentary behaviour and PA intensities for each of the levels of difficulty performed and by total activity. Differences according to sex are also shown. Of the three levels, the Intermediate level was the one with the highest number of minutes of MVPA (3:34 min) for both men (3:45 min) and women (3:21 min). The level with the lowest number of minutes of MVPA (2:26 min) was the highest difficulty level. When the three levels were added together, the number of minutes of MVPA was 08:53. The totals for each participant can be found in the Appendix A. In terms of sex differences, men had higher MVPA levels at all three levels, but these differences were not significant. The only significant difference between the sexes was in favour of men at the moderate intensity of the novice level, with a small effect size.

## 4. Discussion

The aim of the present study was to determine the duration and intensity of physical activity performed using immersive virtual reality, as well as the differences in PA intensity according to sex. The results show that, out of the 25 min of the PA programme, MVPA was performed for 8:53 min. Only at the Beginner level was moderate-intensity PA significantly more common in men.

Previous research has shown that IVR can be a tool to encourage PA practice [24] and that it promotes adherence to PA recommendations [29]. However, few studies have used accelerometers in an IVR PA setting, so our study contributes to increasing the knowledge base on this topic.

Measuring the intensity of PA practice is important because it allows us to determine whether it contributes to health benefits. Several studies have shown that MVPA PA provides the greatest health benefits [2,3]. And because the evidence supports IVR as a valid tool that contributes to physical activity [20], it is necessary to determine the time and intensity that it can lead to.

Analysis of the exercise programme performed shows that the Advanced level produced the lowest MVPA time and the Intermediate level produced the highest. This situation suggests that it is not necessary to perform at the more complex level to obtain higher MVPA values. We believe that this situation is in line with what Tao et al. [43] reported when they stated that IVR activities are a promising tool to address various health-related contexts, but more research is needed on the design of the games to obtain a greater benefit. On the other hand, the fact that the Intermediate level provided the greatest benefit in terms of exercise intensity means that the benefits are not exclusive to people with a high level of skill in using these devices, thus promoting greater enjoyment and adherence to the activity.

As there is still little research on PA intensities in IVR, it is difficult to find studies with which to compare. The closest was reported in a study in which the perceived exertion and PA intensity of different virtual reality games were among the measured variables, where it was found that the game with the highest perceived exertion was the one that elicited the highest levels of MVPA [37]. These discrepancies (i.e., the most complex level in our study did not elicit the highest scores, whereas the research with higher perceived effort elicited higher scores) may be due to the fact that the perceived effort is a subjective measure, and the sample evaluated corresponded to sedentary university students, whose perception may be influenced by them not being used to consistent PA practice rather than because of the higher level of the games. However, due to the paucity of evidence on PA through IVR, further studies are needed to confirm or refute what we have reported.

On the other hand, it is already known that, in the therapeutic field, IVR has great potential to contribute to adherence, as it is considered a tool that enables behavioural change [44,45,46,47]. Therefore, further research is needed to determine, with more scientific support, the contribution of IVR to the field of PA. In this sense, there is already information suggesting that performing PA through IVR provides participants with high levels of enjoyment [20], increased calorie expenditure, improved concentration, and a feeling of being able to continue exercising for longer [46]. This, together with emerging evidence that playing video games in IVR leads to moderate and vigorous PA [38,39], supports the benefits that IVR could provide.

The analysis of total time shows that the average time spent on MVPA was 8:53 min, or 35.5% of the total time (25 min). For an adult, 150–300 min of moderate-intensity physical activity, or 75–150 min of vigorous-intensity physical activity, or an equivalent combination of moderate- and vigorous-intensity physical activity, per week is recommended [6]. To meet the recommendations, it seems necessary to increase the number of sessions or to include other types of PA.

### 4.1. Practical Applications

The practical applications of our results allow us to support the implementation of interventions within universities using IVR not only as a tool to enhance professional training but also as an option to promote PA due to the high percentage of MVPA generated in short sessions. This is relevant, as university students are a population with poor health habits and little time for PA practice due to the demands of their education [48].

### 4.2. Limitations and Strengths

This study has some limitations. The use of only one exercise programme is a limitation, as there is evidence that the types of games used through VR [49,50], as well as their configuration [51,52], produce different levels of effort. In addition, the level of sedentary behaviour and the intensity of physical activity were not measured before performing the physical exercise programme through VR. From the methodological field, the cross-sectional design is also a limitation since it prevents a better understanding of what was found, and working with a non-representative sample prevents generalizing the results. We believe that the characteristics of the sample are also a limitation, as there is considerable evidence to suggest that student athletes have high levels of physical activity [53,54,55]; therefore, the results of this study should be analysed with caution and not extrapolated to most members of the adult population, who have low levels of PA practice [7]. It is therefore imperative that further studies are carried out that include the variables and tools studied here but this time in a sample that has the characteristics of most of the population. A final limitation is the high cost of IVR devices, which has decreased over time but still makes it difficult for the general population to access them. The strengths of the present research lie in the use of an objective instrument to measure PA, applied to a variable with little scientific evidence to date, which provides the first steps for further research.

### 4.3. Future Research Directions

It is recommended that future research should further investigate what has been investigated here by conducting longitudinal studies and, if possible, randomised controlled trials. Due to the increasing use of IVR, not only in research but also for personal use, it is recommended that the intensity of PA be investigated in the wide range of applications available for performing PA using IVR.

As there is already evidence that IVR is a tool for promoting healthy behaviours, it is recommended to investigate the role that IVR could play in promoting adherence to physical activity.

## 5. Conclusions

The results of this study suggest that an immersive virtual reality-based exercise programme produces MVPA levels that do not differ significantly by sex.

This study highlights the need for further research to confirm the contribution of IVR to PA. For a better understanding, longitudinal studies are recommended, possibly in randomised controlled trials, in samples with higher levels of sedentary behaviour, and using several of the available applications to perform PA in IVR.

## Figures and Tables

**Table 1 healthcare-11-02399-t001:** Sedentary behaviour and physical activity intensities by level and total physical exercise programme.

	Total	Women (*n* = 18)	Men (*n* = 21)	*p*	ES
Beginner (8 min)					
Sedentary (min)	01:14	01:24	01:05	0.164	-
Light (min)	03:52	04:11	03:36	0.115	-
Moderate (min)	02:34	02:06	02:58	**0.045**	0.148
Vigorous (min)	00:19	00:18	00:20	0.388	-
MVPA (min)	02:53	02:24	03:18	0.139	-
Intermediate (9 min)					
Sedentary (min)	01:01	00:58	01:04	0.385	-
Light (min)	04:25	04:41	04:11	0.171	-
Moderate (min)	03:07	02:57	03:16	0.286	-
Vigorous (min)	00:27	00:24	00:29	0.331	-
MVPA (min)	03:34	03:21	03:45	0.139	-
Advanced (8 min)					
Sedentary (min)	01:18	01:31	01:08	0.176	-
Light (min)	04:16	04:16	04:16	0.495	-
Moderate (min)	01:56	01:46	02:05	0.253	-
Vigorous (min)	00:30	00:28	00:31	0.357	-
MVPA (min)	02:26	02:14	02:36	0.139	-
TOTAL (25 min)					
Sedentary (min)	03:34	03:53	03:17	0.245	-
Light (min)	12:33	13:08	12:03	0.192	-
Moderate (min)	07:37	06:49	08:19	0.131	-
Vigorous (min)	01:16	01:10	01:21	0.320	-
MVPA (min)	08:53	07:59	09:40	0.139	-

ES effect size (Cohen’s d) MVPA moderate to vigorous physical activity. The values in bold indicate a statistical significance for *p* < 0.05. Game level durations = Beginner: 8 min; Intermediate: 9 min; Advanced: 8 min. Total session: full game session considering the three levels.

## Data Availability

The data presented in this study are available on request from the corresponding author.

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
