# Peer review of "Intensity of a Physical Exercise Programme Executed through Immersive Virtual Reality"

_healthcare, 2023, doi:10.3390/healthcare11172399_

Round 1

Reviewer 1 Report

In this short paper, the authors present their work on using VR exergames to determine the duration and intensity of physical activity and to see if there are differences between males and females. While past research has shown that there is usually not much difference between females and males, the part about physical activity offers some insights into the use of IVR for exercising.

The paper is, in general well, written. Given that it is a brief report, the amount and type of work explored meet the requirements.

Below are some aspects that the authors can clarify or improve further.

 - Provide a summary of findings at the end of Section 1 (Introduction). This will help readers understand the contributions without the need to finish reading the paper.

- Results. Table 1 provides all the main results, but there are limited details on how to read it. Additional descriptions could be provided in the main text or in the caption to guide readers.

- Baseline comparison. I wonder why the authors did not consider a baseline comparison, either having participants do the same type of exercises without involving a computer game or using a normal display TV display, given that research (e.g., https://doi.org/10.1145/3411764.3445801; https://doi.org/10.3390/soc11040134) has found that different types of displays (immersive and non-immersive) could lead to different exertion levels.

 - Qualitative data. Did the authors collect qualitative data about the possible future adoption of these tools? What was the feeling of participants after using the tool?

 In addition, it would be useful to know, given that the authors want to promote IVR if their participants would want to use it later. Also, what features do they like (did not like)? Furthermore,  what factors could constrain them from using IVR in the future? While it is not expected that the authors do a factor analysis such as this (https://doi.org/10.1080/10447318.2022.2098559), it would be helpful to know if there are some main rough factors their participants care (or don't care) about.

 - Limitations. Finally, I think that the authors could delve deeper into their discussions about the limitations, as they have tested only one type of exergame, whereas other types of physical games could require greater exertion (e.g., https://doi.org/10.1145/3383668.3419958; and https://doi.org/10.1145/3544548.3580973) and setups (e.g., https://doi.org/10.3390/soc11040134; https://games.jmir.org/2020/3/e17972/) and can also lead to different activity levels.

Minor aspects:

 - Meta Guest 2 --> Meta Quest 2

 - with a break of xx minutes between levels --> xx? actual numbers are missing.

Author Response

Below are some aspects that the authors can clarify or improve further.

 - Provide a summary of findings at the end of Section 1 (Introduction). This will help readers understand the contributions without the need to finish reading the paper.

A: Thank you very much for your comment. We do not add what is suggested, since in the introduction we incorporate the background and the objective of the research. The results found are indicated in the summary, which helps the reader understand what this research contributes.

- Results. Table 1 provides all the main results, but there are limited details on how to read it. Additional descriptions could be provided in the main text or in the caption to guide readers.

A: Thank you for your comment. We have added more detail in the text and in the table. In addition, as a suggestion from another reviewer, we have added as supplementary material the average results obtained by each of the 39 evaluated.

- Baseline comparison. I wonder why the authors did not consider a baseline comparison, either having participants do the same type of exercises without involving a computer game or using a normal display TV display, given that research (e.g., https://doi.org/10.1145/3411764.3445801; https://doi.org/10.3390/soc11040134) has found that different types of displays (immersive and non-immersive) could lead to different exertion levels.

A: We appreciate your comment. Because the laboratory where the virtual reality instruments are located is recently implemented (March 2023), and because this research is the first step we are taking in evaluating the amount of physical activity through virtual reality, we do not We consider as an objective to compare with some previous reference, or using different screens (immersive and non-immersive).

The bibliographic search carried out for the writing of this research allowed us to identify evidence that reports differences according to the use of different programs or screens. Situation that is corroborated with the comments exposed in the review that has been made to us. For these reasons, we have already considered collecting data in the future where comparisons are made between protocols or with the use of different devices.

 - Qualitative data. Did the authors collect qualitative data about the possible future adoption of these tools? What was the feeling of participants after using the tool?

A: Thank you for your comment. We do not collect qualitative data. The feeling of the participants at the end of the session was good. Most of them gave opinions "that it was an entertaining activity", "that it was a novel activity", among others. And many of them repeated it another time just for the pleasure of practicing it.

Unfortunately, we realized late about the importance of the qualitative data provided by the participants, which is why they were not collected.

However, it is already within the future investigations that we will carry out using reality. In the future they will be compiled.

 In addition, it would be useful to know, given that the authors want to promote IVR if their participants would want to use it later. Also, what features do they like (did not like)? Furthermore,  what factors could constrain them from using IVR in the future? While it is not expected that the authors do a factor analysis such as this (https://doi.org/10.1080/10447318.2022.2098559), it would be helpful to know if there are some main rough factors their participants care (or don't care) about.

A: Different studies analyze the acceptance and positive vision in the use of this technological tool, however, research on the objective measurement of the intensity of physical activity through immersive virtual reality is still incipient. For this reason, the research group decided to define this purpose and in future research, expand the work that enhances work in this area.

 - Limitations. Finally, I think that the authors could delve deeper into their discussions about the limitations, as they have tested only one type of exergame, whereas other types of physical games could require greater exertion (e.g., https://doi.org/10.1145/3383668.3419958; and https://doi.org/10.1145/3544548.3580973) and setups (e.g., https://doi.org/10.3390/soc11040134; https://games.jmir.org/2020/3/e17972/) and can also lead to different activity levels.

A: Thank you very much for your comment. As recommended, we have added information on the limitations, using the references provided.

Minor aspects:

 - Meta Guest 2 --> Meta Quest 2

 - with a break of xx minutes between levels --> xx? actual numbers are missing.

A: Modified

Reviewer 2 Report

I would suggest that you give more details on the group as well as on the procedure. In the result session there is also improvement for some more details. The cohort is not that large. Therefore I suggest to show some individual results, too. 

In my opinion this is a good paper. A native speaker should have a deeper look into grammar. 

Author Response

I would suggest that you give more details on the group as well as on the procedure. In the result session there is also improvement for some more details. The cohort is not that large. Therefore I suggest to show some individual results, too. 

A: We appreciate your review. More details of the group evaluated, the procedure carried out and the results have been added. As suggested, we have added the values obtained by each of the participants in each of the executed levels, which can be consulted in supplementary material.

In my opinion this is a good paper. A native speaker should have a deeper look into grammar. 

A: A native speaker has reviewed the manuscript.

Reviewer 3 Report

Thank you for submitting this review entitled 

“Intensity of a physical exercise program executed through immersive virtual reality”

I am going to comment on some doubts, suggestions that the authors must take into account to facilitate reading for the reader.

-          Affiliations must follow the same guidelines and in English.

-          Summary conclusions are the conclusions drawn from your results, not the need to do research on the topic.

-          The references in the text do not match the numbers in the bibliography. The authors must check all the references of the text and that they coincide with the bibliography.

-          Regarding material and methods, it is interesting that they say what the cut-off point is for a subject to be considered sedentary, in addition to referencing the study by the author Arias Palencia et al.

-          It would have been interesting to see what level of physical activity or sedentary lifestyle the participants present at baseline, since in one of their objectives they want to determine differences in the intensity of physical activity according to gender. Since it seems that they have not done it, they should comment on it as a limitation of their study.

-          In the procedure they do not indicate the rest time between one training session and another “Each level lasted between 8 and 9 minutes, with a break of xx minutes between levels.”

-          In the discussion, they indicate that "Only at the novice level was moderate intensity significantly higher in men", according to the table they would be women.

-          As conclusions they indicate that:

The results of our study indicate that the exercise program delivered via IVR generates MVPA (8:53 out of 25 minutes), with no major gender differences.

The first part is not a conclusion, it is a result. Please indicate what the values obtained mean.

Thanks for the work, I hope you can resolve these issues.

Author Response

I am going to comment on some doubts, suggestions that the authors must take into account to facilitate reading for the reader.

-          Affiliations must follow the same guidelines and in English.

A: Thank you for your comment. We have unified the writing of the affiliations of all the authors. The address (Lota 2465) was added to author 3, since his institution requires it.

With regard to writing all affiliations in English, it is not possible since the institutions where we work require us to do so in Spanish.

-          Summary conclusions are the conclusions drawn from your results, not the need to do research on the topic.

A: Thank you for your comment. As suggested, we have modified the conclusions described in abstract.

-          The references in the text do not match the numbers in the bibliography. The authors must check all the references of the text and that they coincide with the bibliography.

A: We appreciate your comment. We have corrected the errors found in the references.

-          Regarding material and methods, it is interesting that they say what the cut-off point is for a subject to be considered sedentary, in addition to referencing the study by the author Arias Palencia et al.

A: Thank you for your comment. We have added the requested information.

-          It would have been interesting to see what level of physical activity or sedentary lifestyle the participants present at baseline, since in one of their objectives they want to determine differences in the intensity of physical activity according to gender. Since it seems that they have not done it, they should comment on it as a limitation of their study.

A: Thank you very much for your comment. We have added the suggested limitation.

-          In the procedure they do not indicate the rest time between one training session and another “Each level lasted between 8 and 9 minutes, with a break of xx minutes between levels.”

A: Modified.

-          In the discussion, they indicate that "Only at the novice level was moderate intensity significantly higher in men", according to the table they would be women.

A: We have carefully reviewed the results, and what we have written is correct (Only at the beginner level was moderate intensity significantly higher in men), since women present an average of 02:06 minutes in moderate intensity, while men present 02:58.

-          As conclusions they indicate that:

The results of our study indicate that the exercise program delivered via IVR generates MVPA (8:53 out of 25 minutes), with no major gender differences.

The first part is not a conclusion, it is a result. Please indicate what the values obtained mean.

A: Thank you for your comment. According to the recommendations, we have modified the first part of the conclusions. 

Thanks for the work, I hope you can resolve these issues.

Round 2

Reviewer 1 Report

I thank the authors for their responses and for revising their paper based on the comments from the first round. Overall, the paper's readability has improved--it is now clearer and complete. Congratulations on the work done.